# Six-month psychopathological symptom trajectories following the COVID-19 outbreak: Contrasting mental health outcomes between nurses and the general population

**Catarina Vitorino** *, **Maria Cristina Canavarro, Carlos Carona**

Center for Research in Neuropsychology and Cognitive-Behavioral Intervention, Faculty of Psychology and Educational Sciences, University of Coimbra, Coimbra, Portugal

* catarina.alves.vitorino@gmail.com

## Abstract

The COVID-19 pandemic prompted a social, economic and health crisis that had a major impact on the mental health of the global community, particularly nurses. The objective of the current study is to conduct a longitudinal evaluation of the trajectory of depressive, anxiety, trauma, and fear of COVID-19 symptoms, comparing self-reports of nurses and the general population over a six-month period. Self-report questionnaires were administered online to a sample of 180 nurses and 158 individuals from the general population for the baseline assessment (T1) and follow-up at 6 months (T2). Levels of symptoms reported by nurses were generally greater and tended to worsen over time, as opposed to the levels of symptoms reported by the general population that tended to improve. Levels of depressive, anxiety, and trauma symptoms were significantly different between nurses and the general population over time. Levels of fear of COVID-19 declined significantly from T1 to T2 in both groups. These results suggest that it is crucial to monitor the longer-term effects of COVID-19 and to develop resilience-promoting interventions tailored to the unique needs of this vulnerable group.

**Data Availability Statement:** All relevant data and reported findings will be shared with other members of our Research Center who did not

## Introduction

The COVID-19 outbreak spread worldwide in March 2020, and since then, it has had a tremendous impact on people's daily lives and well-being. Beyond the major concerns about one's physical health and that of significant others, which are both primary sources of stress, the consequences of COVID-19 have been widely generalized to psychological, social, and economic dimensions of life [1]. Studies about the impact of the pandemic and its short- and long-term effects, across a wide range of mental health outcomes and populations, are growing in number [2–5]. However, little is known about the evolution of psychological symptoms at various stages of the public health emergency, as well as which groups present a higher risk of psychological distress.

collaborate with the present study to ensure long-term storage and access. As such, external data requests may be sent to the Center for Research in Neuropsychology and Cognitive-Behavioral Intervention by email (cineicc@fpce.uc.pt), in case the corresponding author becomes unavailable.

**Funding:** This study was supported by a doctoral grant (2020.07981.BD) awarded to C. V., the first author (Portuguese Foundation for Science and Technology/MCTES - https://www.fct.pt/en/), and by the Center for Research in Neuropsychology and Cognitive–Behavioral Intervention (CINEICC - https://cineicc.pt/) at the University of Coimbra (UIDB/PSI/00730/2020) for C. C. The APC was funded by the Center for Research in Neuropsychology and Cognitive-Behavioral Intervention (CINEICC) at the University of Coimbra. The funders had no role in study design, data collection and analysis, decision to publish, or preparation of the manuscript.

**Competing interests:** The authors have declared that no competing interests exist.

In the general population, cross-sectional studies around the globe have shown elevated rates of distress and posttraumatic stress, anxiety and depressive symptoms during the early stages of the pandemic [6, 7]. Although it has had an impact on the entire global community, there are specific social groups that have been more exposed to the deleterious consequences of COVID-19 [8]. Health care workers, especially nurses, were one of those at-risk groups due to their occupational role. Without adequate resources to perform their jobs, they were expected to be present during every stage of the pandemic to directly treat, care for, and support patients. Therefore, they experienced an increased risk of developing pandemic-related stress [9]. According to a meta-analysis conducted between January 2020 and September 2020, approximately one-third of nurses have experienced stress, anxiety, depression, and sleep disturbance, which represents higher levels than those described by the general population during the same stage of infection, as well as by nurses during previous infectious diseases (e.g., MERS and SARS epidemics) [10]. Nonetheless, the results have not been conclusive about the psychosocial effects of the COVID-19 pandemic, particularly when comparing both groups [11, 12].

The continuous cumulative nature of COVID-19 traumatic stressors, along with the lack of an end in sight, has been shown to predict severe and enduring mental health problems [1]. Although scarce in the literature, longitudinal studies are crucial to understand the course of symptoms over time and to identify the pathways through which individuals adjust to traumatic events. A longitudinal evaluation of the psychological impact of the COVID-19 pandemic in the Spanish population found a significant increase in depression, anxiety, and stress scores at the beginning of the outbreak but no significant temporal changes in the psychological impact of the event over a period of one month [13]. Another longitudinal study conducted in Argentina reported an increase in the levels of depression, while symptoms of anxiety and both positive and negative affect showed a decrease over the first 2 weeks after the quarantine was imposed [14]. In the UK, the impact of fear of COVID-19 significantly decreased between the time of the highest number of pandemic-related deaths and the use of vaccine and subsequent decline of victims, which was separated by four months [15]. A recent systematic review and meta-analysis of longitudinal cohort studies on mental health problems before and during the pandemic concluded that psychological symptoms increased soon after the COVID-19 outbreak but then decreased to levels comparable to those in the prepandemic period [16]. Even though the body of literature is increasing, the evidence available on the long-term effects of the pandemic is still insufficient and incomplete.

While the whole community has faced different levels of confinement, social restrictions, job loss, and institutional closures, health care professionals were pushed to their limits. Those who were in direct contact with patients, such as nurses, were one of the groups most affected by the pandemic. This public health crisis presented additional pressure and challenges to the nursing workforce, exacerbating already difficult working conditions prior to the pandemic. During the COVID-19 pandemic, empirical evidence demonstrated that nurses reported greater levels of depression, anxiety, and posttraumatic stress symptoms during the peak of the outbreak than during the stable stage of the pandemic [17]. Another longitudinal study administered in the UK during the first wave of the pandemic reported that nurses experienced a high prevalence of negative psychological effects, including severe stress and anxiety, and demonstrated high scores on the impact of event, indicative of probable posttraumatic stress disorder [18]. A longitudinal qualitative interview study identified the anxiety, frustration, guilt, and inner turmoil conveyed by the UK nurses' narratives [19]. However, fear of COVID-19 has not been longitudinally assessed as a core outcome in these professionals. The main function of fear is to protect an individual from an immediate known external danger (COVID-19, in this case) by eliciting the expression of a range of adaptive and defensive behaviors [20]. Exploring this emotional response in the specific context of COVID-19 is of paramount

importance to understand the impact, not only on compliance with safety measures to contend with the spread of the disease but also on the adaptation process to this global health crisis (i.e., functional and adaptive versus irrational and pathological fear responses) [15].

The results from longitudinal studies are still not consistent about the trajectory of symptoms. The methodological and ecological characteristics of the study may influence the nature of the change that is found, such as the timepoint at which the assessments were performed, the local measures that were taken, the aspect of the mental health that is analyzed, and the kind of job involved (e.g., working directly with COVID-19 patients versus nonfrontline work) [21, 22]. Therefore, there is an urgent need to collect high-quality data in the postpandemic period on prevalence rates of depression, anxiety and other mental health effects across the whole population and vulnerable groups, including nurses [23, 24]. Contrasting mental health outcomes between samples enables the development of evidence-based guidelines on how to respond to this and future pandemics or infection waves and tailor prevention and intervention according to the needs of specific at-risk groups.

## The current study

The present study was conducted in a sample of nurses and individuals from the general population to compare the stability/change in the adaptation outcomes between both groups at two timepoints separated by 6 months. As a recent and enduring global health crisis, the short-term effects of the COVID-19 pandemic on mental health have been largely investigated across countries [12, 25–29]. However, findings are mixed and inconsistent due to the instability that has characterized this pandemic. As a result, longitudinal evidence is crucial to better understand the trajectory of symptoms from the peak of the outbreak to the ease of lockdown measures. Furthermore, the intensification of global disasters in the past couple of decades requires coordinated public health approaches to develop tailored interventions with an evidence-based framework to mitigate adverse responses to trauma [30, 31]. Collecting high-quality data about the prevalence of psychological symptoms among different groups, contrasting the adaptation outcomes and investigating specific needs can provide a robust understanding of the impact of current and future mass traumas on the public's mental health.

Given the abundant participation requests in postdisaster research, low response rates have been frequently indicated as a major limitation of longitudinal studies in the context of the COVID-19 pandemic [32]. To prevent heightened attrition rates, a short timeframe between assessments was used to produce more reliable results and a consistent influence on the results and the subsequent response to the public health emergency.

The main goal of the current study was to longitudinally assess the levels of depressive, anxiety, trauma and fear of COVID-19 symptoms in nurses and the general population over a six-month period.

## Materials and methods

### Participants and procedures

The present study is included in a broader research project that intends to explore the psychosocial implications of the COVID-19 outbreak on Portuguese nurses in comparison with the general population. It involves a retrospective, longitudinal design with two assessment timepoints: baseline (T1) and follow-up at 6 months (T2).

All procedures were in accordance with the Declaration of Helsinki and its later amendments for research involving human participants (World Medical Association, 2013). The study was approved by The Ethics Committee of the Faculty of Psychology and Educational Sciences of the University of Coimbra.

The research sample was invited to participate in this study through social and traditional media platforms and institutional email lists by means of unpaid cross-posting, paid advertisements and booster campaigns. To facilitate the dissemination of the study, the Portuguese Order of Nurses, Portuguese Nurses Unions and the Nursing Schools of Coimbra and Lisbon approved and celebrated a partnership protocol. The two inclusion criteria for the community sample to participate in this study were: being older than 18; and being able to understand Portuguese and living in Portugal. To answer the survey specific for the group of nurses, there were a single inclusion criterion: being a nurse working in a Portuguese hospital or any other health care institution.

The baseline assessment (T1) was conducted between September 2021 and May 2022, while the second point of data collection (T2) was performed between May 2022 and December 2022. The first page of the survey provided information about the study aims and procedures, as well as the voluntary, anonymous, and confidential nature of the investigation. All participants had to provide an informed consent to participate in the study, by clicking on the option "I understand and accept the conditions of the study". They completed the Portuguese validated versions of the self-report instruments, using a web-based platform (LimeSurvey®).

Respondents who consented to provide an e-mail contact received an invitation to voluntarily participate again in the study at the 6-month follow-up. To prevent missing values, forced answering (i.e., forcing respondents to answer each question to proceed through the questionnaire) was used, except for the e-mail contact. Of the 1335 participants (672 nurses and 663 individuals from the general population) on the baseline assessment, 338 completed the follow-up (25.3%), with a mean age of 38.4 ($SD$ = 10.58), 89.3% ($n$ = 302) women, 10.1% ($n$ = 34) men, and 0.2% ($n$ = 2) nonbinary individuals. Thus, the attrition rate from T1 to T2 was 75% for the total sample, which is in line with previous reports on postdisaster research [32, 33]. For the group of nurses, the attrition rate was 73%; for the general population, it was 76%.

## Measures

**Sociodemographic and clinical information.**   A sociodemographic and clinical questionnaire was created specifically for this study, to the baseline assessment, and included questions about sociodemographic (e.g., age, gender, marital status, residence) and health-related data (e.g., psychologi-cal/psychiatric treatment history). The level of exposure to COVID-19 was measured by asking about the intensity of contact with the SARS-CoV-2 virus, such as a positive diagnosis of COVID-19 (of oneself or significant others), as well as the period and intensity of isolation (e.g., "Were you in isolation?", "Were you infected with the coronavirus?", "Were there any of your friends infected with the coronavirus?"). Nurses also gave information about years of professional experience and whether they worked in a COVID-19 unit during the outbreak of the pandemic.

**Depressive symptoms.**   The Overall Depression Severity and Impairment Scale (ODSIS) [34, 35] evaluates the severity of depressive symptoms and the resulting interference on daily routine. This self-report questionnaire, developed by Bentley and colleagues (2014), contains 5 questions (e.g., "When you feel depressed, how intense or severe is your anxiety?", "How often do you avoid situations, places, objects, or activities because of depression?") and the answers ranged from 0 = *None* to 4 = *Extreme*, referring to "the past week". The overall score is created through the sum of all items and can range from 0 to 20, with higher scores suggesting more frequent and severe depressive symptoms. The recommended cutoff score for the original version was 8 [34]. The reliability of the scale was high within the current study (total sample: T1: $\alpha$ = .94; T2: $\alpha$ = .95; general population: T1: $\alpha$ = .95; T2: $\alpha$ = .94; nurses: T1: $\alpha$ = .94; T2: $\alpha$ = .95).

**Anxiety symptoms.**    The Overall Anxiety Severity and Impairment Scale (OASIS) [36, 37] measures anxiety-related severity and functional impairment. This self-report questionnaire, created by Norman and colleagues (2006), contains 5 questions (e.g., "How often have you felt anxious?", "How much has anxiety interfered with your social life and relationships?") rated on a 5-point Likert scale (from 0 = *None* to 4 = *Extreme*) and the timeframe it refers to is in "in the past week". The total score can range from 0 to 20, and it is calculated by adding the score of each item, with higher scores indicating more frequent and severe anxiety symptoms. The cutoff score of the original version was 8 [36]. The OASIS showed excellent internal consistency in the present study (total sample: T1: $\alpha$ = .92; T2: $\alpha$ = .93; general population: T1: $\alpha$ = .92; T2: $\alpha$ = .92; nurses: T1: $\alpha$ = .92; T2: $\alpha$ = .94).

**Trauma symptoms.**    The Impact of Event Scale-6 (IES-6) [38, 39] is a brief version used to assess posttraumatic stress symptoms that was designed by Thoresen and colleagues (2010). The IES-6 consists of 6 items (e.g., "I thought about it when I didn't mean to.", "I was aware that I still had a lot of feelings about it, but I didn't deal with them.") rated on a 5-point Likert scale (from 0 = *Never* to 4 = *Extremely*). The total score can range from 0 to 24, and it is calculated by the sum of all items, with higher scores indicating higher levels of trauma symptoms. The cutoff score of 12.5 was indicated for clinically significant trauma symptoms in the Portuguese population [38]. In this sample, the IES-6 revealed very good internal consistency for the total score (total sample: T1: $\alpha$ = .86; T2: $\alpha$ = .88; general population: T1: $\alpha$ = .86; T2: $\alpha$ = .85; nurses: T1: $\alpha$ = .85; T2: $\alpha$ = .88).

**Fear of COVID-19.**    The Fear of COVID-19 Scale [40, 41] evaluates individual fear of coronavirus and it was produced by Ahorsu and colleagues (2019). It is composed of 7 items (e.g., "It makes me uncomfortable thinking about COVID-19.", and "I am afraid of losing my life because of COVID-19.") rated on a 5-point Likert scale (1 = *Strongly disagree* to 5 = *Strongly agree*). The total score ranges from 7 to 35 and is calculated by adding up all items, with higher scores corresponding to greater fear of the pandemic. According to the Greek version, participants scoring $\geq$ 16.5 were categorized as having extreme fear of COVID-19 [42]. Cronbach's alphas for the scale were high within the current study (total sample: T1: $\alpha$ = .93; T2: $\alpha$ = .89; general population: T1: $\alpha$ = .87; T2: $\alpha$ = .88; nurses: T1: $\alpha$ = .88; T2: $\alpha$ = .89).

Although no permission is needed to use the scale, the original authors were informed about the translation of the questionnaire to Portuguese through a forward-backward translation procedure. First, the scale was translated independently by two researchers (A. Fonseca & M. M. Ramos) who were fluent in Portuguese and English and familiar with the concepts included in the questionnaire. After that, a third person, who was unfamiliar with the scale, was responsible for back-translation of the Portuguese version into English. Finally, both versions (original and back-translation) were compared to achieve a harmonized and conceptually coherent Portuguese version.

## Data analysis

Statistical Package for the Social Sciences (SPSS, version 27.0; IBM SPSS, Chicago, IL, USA) was the software selected to perform data analyses. Based on a priori power analysis (G*Power) [43] to detect medium-to-large effects in planned statistics (i.e., correlational analyses, repeated measures, within-between interactions), a minimum of 195 individuals was needed. Cronbach's alphas informed about the internal consistency of the questionnaires, considering $\alpha \geq$ .80 as optimal [44]. Descriptive statistics were obtained for all variables under study, and differences in sociodemographic and clinical variables were tested through mean differences tests (Student's t tests) or frequency differences for categorical variables (chi-square tests). For descriptive purposes, two groups were created for each subsample according to the

recommended cutoff points on the total score of the ODSIS [34], OASIS [36], IES-6 [38], and Fear of COVID-19 Scale [42]: (1) Group with subclinical symptoms; (2) Group with clinical symptoms.

Pearson's bivariate correlation coefficients were computed to assess associations between depressive, anxiety, and trauma symptoms and fear of COVID-19 for the baseline assessment (T1) and follow-up (T2), while adopting the following guidelines to classify their strength: $r \leq$ .29 (weak); .30 $\leq r \leq$ .49 (moderate); $r \geq$ .50 (strong) [45].

A mixed model ANOVA was used to assess the effects of group and time on individuals' mental health outcomes in the aftermath of the COVID-19 pandemic (depressive, anxiety, and trauma symptoms and fear of COVID-19). Group was considered the intersubject factor, and time point was considered the intrasubject factor. Preliminary assumption testing was conducted to check for normality, linearity, univariate and multivariate outliers, homogeneity of variance-covariance matrices, and multicollinearity [46]. No serious violations were noted, except for the homogeneity of variance of anxiety symptoms at T2. Following multivariate effects, univariate analyses were performed using Bonferroni correction not only to control alpha inflation owing to multiple testing but also to compensate for the violation of the assumption on the homogeneity of variances [47]. Significant interaction effects between group and time were explored using simple effects tests, comparing the effect of the group at each time and the effect of time for each group. The statistical significance of the indirect effects was tested using a bootstrapping procedure with 5000 samples, which generated 95% bias-corrected and accelerated confidence intervals (95% BCaCIs) [47]. When the value of zero was not contained in the confidence intervals, the indirect effect was significant. For the comparison analyses, effect-size measures (partial eta squared) were presented considering $\eta_P^2$ = 0.01 as a small effect, $\eta_P^2$ = 0.06 as a medium effect and $\eta_P^2$ = 0.14 as a large effect [48]. Furthermore, it was depicted a graphical illustration of the effects.

A minimum confidence interval of 95% was considered for all the analyses performed in this study.

## Results

### Sociodemographic and clinical characteristics of the sample

Table 1 summarizes the sociodemographic and clinical characteristics of the subsamples. The general population group included 158 participants (age: $M$ = 37.8; $SD$ = 11.8), while the group of nurses consisted of 180 individuals (age: $M$ = 38.9; $SD$ = 9.36). For both groups, the majority identified as women (87.3% for the general population, 91.1% for nurses), married (47.5% for the general population, 53.9% for nurses) and urban residents (75.3% for the general population, 76.7% for nurses). Being in isolation and integrating the risk group for COVID-19 were the only variables that showed statistically significant differences between both groups.

Concerning the group of nurses, the mean years of experience was 15.7 ($SD$ = 9.34), and the majority worked in a COVID-19 unit (95%) during the pandemic outbreak. As for the general population, most individuals had an academic degree (47.5%) and worked on-site (53.2%).

Regarding the clinical levels of symptoms presented in Table 2, nurses reported higher levels than the general population for all measures, ranging between 22.8% and 31.7% at T1, and 21.7% and 43.3% at T2. Among the general population, symptoms with clinical significance ranged between 15.8% and 29.1% at T1, 11.4% and 24.7% at T2. Differences between time-points were statistically significant for both groups on all measures; differences between groups were significant only at the second moment of assessment for depressive, anxiety and trauma symptoms.

**Table 1. Sociodemographic and clinical characteristics.**

| | Nurses (N = 180) | General population (N = 158) | Differences between samples |
|---|---|---|---|
| Age (M/SD) | 38.9 (9.36) | 37.8 (11.8) | $t = -.95; p = .34$ |
| Gender (n/%) | | | |
| Woman | 164 (91.1) | 138 (87.3) | $x^2 = 1.28; p = .53$ |
| Man | 15 (8.3) | 19 (12) | |
| Nonbinary | 1 (.6) | 1 (.6) | |
| Marital status (n/%) | | | |
| Single | 73 (40.6) | 70 (44.3) | $x^2 = 4.66; p = .20$ |
| Married | 97 (53.9) | 75 (47.5) | |
| Divorced | 8 (4.4) | 13 (8.2) | |
| Widow | 2 (1.1) | 0 (0) | |
| Residential area (n/%) | | | |
| Urban | 138 (76.7) | 119 (75.3) | $x^2 = .08; p = .78$ |
| Rural | 42 (23.3) | 39 (24.7) | |
| Isolation (n/%) | | | |
| Yes | 61 (33.9) | 77 (48.7) | $x^2 = 7.68; p < .01$ |
| Psychological/psychiatric treatment history (n/%) | | | |
| Yes | 86 (47.8) | 80 (50.6) | $x^2 = .27; p = .60$ |
| Risk group for COVID-19 (n/%) | | | |
| Yes | 48 (26.7) | 18 (11.4) | $x^2 = 12.49; p < .001$ |
| Infection with the coronavirus (n/%) | | | |
| Yes | 51 (28.3) | 49 (31) | $x^2 = .29; p = .60$ |
| Years of experience (M/SD) | 15.7 (9.34) | - | - |
| Worked in a COVID-19 unit (n/%) | | | |
| Yes | 95 (52.8) | - | - |

**Table 2. Clinical levels of depressive, anxiety trauma and fear of COVID-19 symptoms among nurses and the general population at T1 and T2.**

| | | Nurses (N = 180) | General population (N = 158) | Differences between samples |
|---|---|---|---|---|
| Clinical depressive symptoms (n/%) | | | | |
| Yes * | T1 | 53 (29.4) | 40 (25.3) | $x^2 = .72; p = .4$ |
| | T2 | 72 (40) | 38 (24.1) | $x^2 = 9.75; p < .01$ |
| Differences between timepoints | | $x^2 = 31.45; p < .001$ | $x^2 = 23.73; p < .001$ | |
| Clinical anxiety symptoms (n/%) | | | | |
| Yes * | T1 | 53 (29.4) | 44 (27.8) | $x^2 = .11; p = .75$ |
| | T2 | 78 (43.3) | 39 (24.7) | $x^2 = 12.93; p < .001$ |
| Differences between timepoints | | $x^2 = 28; p < .001$ | $x^2 = 55.75; p < .001$ | |
| Clinical trauma symptoms (n/%) | | | | |
| Yes * | T1 | 41 (22.8) | 25 (15.8) | $x^2 = 2.59; p = .11$ |
| | T2 | 39 (21.7) | 18 (11.4) | $x^2 = 6.34; p < .05$ |
| Differences between timepoints | | $x^2 = 32.02; p < .001$ | $x^2 = 17.82; p < .001$ | |
| Clinical fear of COVID-19 (n/%) | | | | |
| Yes * | T1 | 57 (31.7) | 46 (29.1) | $x^2 = .26; p = .61$ |
| | T2 | 42 (23.3) | 31 (19.6) | $x^2 = .69; p = .41$ |
| Differences between timepoints | | $x^2 = 50.19; p < .001$ | $x^2 = 38; p < .001$ | |

* Group with clinical symptoms (categorized into two groups based on the recommended cutoff points)

**Table 3. Matrix of intercorrelations among study variables.**

|  |  | Time 1 | | | | Time 2 | | |
|---|---|---|---|---|---|---|---|---|
|  |  | **DS** | **AS** | **TS** | **FC** | **DS** | **AS** | **TS** |
| **Time 1** | 1. DS | - | - | - | - | | | |
|  | 2. AS | .80*/.73*/.76* | - | - | - | | | |
|  | 3. TS | .67*/.52*/.60* | .62*/.56*/.58* | .- | .- | | | |
|  | 4. FC | .31*/.42*/.36* | .38*/.42*/.40* | .52*/.55*/.53* | - | | | |
| **Time 2** | 1. DS | .56*/.58*/.57* | .55*/.53*/.52* | .53*/.38*/.48* | .28*/.37*/.32* | - | - | - |
|  | 2. AS | .56*/.55*/.55* | .61*/.67*/.62* | .53*/.47*/.51* | .29*/.36*/.32* | .78*/.77*/.78* | - | - |
|  | 3. TS | .50*/.45*/.47* | .49**/.40*/.43* | .61*/.62*/.62* | .38*/.40*/.37* | .65*/.56*/.64* | .63*/.52*/.60* | - |
|  | 4. FC | .20*/.23*/.22* | .26**/.31*/.28* | .40*/.42*/.42* | .65*/.68*/.66* | .25*/.32*/.29* | .28*/.31*/.30* | .47*/.52*/.49* |

*Notes*: Matrix of intercorrelations for Nurses/General population/Global sample

\* $p < .001$

DS: depressive symptoms, AS: anxiety symptoms, TS: trauma symptoms, FC: fear of COVID-19

## Correlations between depressive, anxiety, and trauma symptoms and fear of COVID-19 for time 1 and time 2

As illustrated in Table 3, all correlations were positive and statistically significant. For the global sample, the strength of the associations was moderate to strong for all variables, except for the weak correlations of fear of COVID-19 (T2) with depressive symptoms (T1 and T2) and with anxiety symptoms (T1). Table 3 describes the intercorrelations between the variables under study for each group, as well as for the global sample.

## Comparison of depressive, anxiety, trauma, and fear of COVID-19 between nurses and the general population over time

Descriptive statistics of study variables are presented in Table 4, and Group, Time, and interaction effects (Group x Time) are described in Table 5. Multivariate tests on mental health outcomes showed statistically significant differences with a large effect size for Time ($F (4, 333) = 15, p < .001, \eta_P^2 = .15$). Separate univariate analysis, using a Bonferroni adjusted alpha level of .01, only revealed significant main effects of Time on the fear of COVID-19 ($F (1, 336) = 55.18, p < .001, \eta_P^2 = .14$).

There was a medium effect size of Group ($F (4, 333) = 8.9, p < .001, \eta_P^2 = .1$) and interaction ($F (4, 333) = 4.7, p < .001, \eta_P^2 = .05$) on mental health outcomes. When the results were analyzed separately, Group main effect was significant for depressive and trauma symptoms ($F (1, 336) = 10.4, p < .01, \eta_P^2 = .03; F (1, 336) = 22.97, p < .001, \eta_P^2 = .06$, respectively), while the main effect of interaction was significant for all measures, except for fear of COVID-19

**Table 4. Descriptive statistics of study variables.**

|  | Nurses (n = 180) | | General population (n = 158) | |
|---|---|---|---|---|
|  | **T1** | **T2** | **T1** | **T2** |
| Depressive symptoms | 5.21 (4.57) | 6.28 (4.86) | 4.57 (4.80) | 4.02 (4.37) |
| Anxiety symptoms | 5.56 (4.31) | 6.39 (4.65) | 5.67 (4.54) | 5.06 (4.20) |
| Trauma symptoms | 8.78 (5.02) | 8.96 (5.17) | 7.20 (5.01) | 5.86 (4.69) |
| Fear of COVID-19 | 14.28 (5.61) | 12.76 (5.44) | 27.97 (5.50) | 11.95 (5.13) |

**Table 5. Group, time, and interaction effects on psychological adjustment to COVID-19.**

| | Group | Time | Group x Time |
|---|---|---|---|
| Multivariate tests | $F(4, 333) = 8.9^{**}$, $\eta_P^2 = .1$ | $F(4, 333) = 15^{**}$, $\eta_P^2 = .15$ | $F(4, 333) = 4.7^{**}$, $\eta_P^2 = .05$ |
| Univariate tests | | | |
| Depressive symptoms | $F(1, 336) = 10.4^{*}$, $\eta_P^2 = .03$ | $F(1, 336) = 1.24$, $\eta_P^2 = .004$ | $F(1, 336) = 11.9^{**}$, $\eta_P^2 = .034$ |
| Anxiety symptoms | $F(1, 336) = 1.9$, $\eta_P^2 = .006$ | $F(1, 336) = .28$, $\eta_P^2 = .001$ | $F(1, 336) = 12.2^{**}$, $\eta_P^2 = .035$ |
| Trauma symptoms | $F(1, 336) = 22.97^{**}$, $\eta_P^2 = .06$ | $F(1, 336) = 5.87$, $\eta_P^2 = .02$ | $F(1, 336) = 10.17^{*}$, $\eta_P^2 = .029$ |
| Fear of COVID-19 | $F(1, 336) = .95$, $\eta_P^2 = .003$ | $F(1, 336) = 55.18^{**}$, $\eta_P^2 = .14$ | $F(1, 336) = 1.39$, $\eta_P^2 = .004$ |

* p < .01

** p < .001

(depressive symptoms: $F(1, 336) = 11.9$, $p < .001$, $\eta_P^2 = .034$; anxiety symptoms: $F(1, 336) = 12.2$, $p < .001$, $\eta_P^2 = .035$; trauma symptoms: $F(1, 336) = 10.17$, $p < .01$, $\eta_P^2 = .029$).

Thus, levels of fear of COVID-19 largely varied from T1 to T2. Nurses' reports of levels of depressive and trauma symptoms were moderately different from those of the general population. Overall, differences between nurses and the general population in depressive, anxiety, and trauma symptoms over time were moderate.

Simple effects analysis comparing the effect of group on time for each measure is depicted in Fig 1. Nurses presented higher levels of symptoms than the general population at both T1 and T2, except for lower anxiety symptoms at T1. In addition, symptoms tended to increase over time for nurses, apart from the decrease found for fear of COVID-19, while the general population demonstrated an improvement in all mental health outcomes.

Levels of depressive symptoms were significantly higher for nurses than for the general population (both at T1 and T2), and the increase in symptoms shown by nurses from T1 to T2 was also larger than the decrease reported by the general population (Fig 1A). The general population revealed higher levels of anxiety symptoms than nurses at T1; while the first group demonstrated a reduction over time, nurses experienced an increase from T1 to T2 (Fig 1B). Trauma symptoms were also significantly higher for nurses at both T1 and T2 than those reported by the general population; the decrease in symptoms shown by the general population over time was larger than the growth reported by nurses (Fig 1C). Regarding fear of COVID-19, both groups showed a significant decrease in symptomatology between T1 and T2 (Fig 1D).

## Discussion

The effects of the COVID-19 pandemic on mental health and well-being have already been studied for a long time. However, it is crucial to understand the magnitude of the impact of this global health crisis, as well as the trajectory of adaptation. This study aimed to explore the differences in the longitudinal evolution of mental health outcomes between nurses and the general population over a six-month period. The main findings may be summarized as follows: first, nurses generally reported higher levels of symptoms than the general population; second, the mental health outcomes of nurses tended to deteriorate over time, except for fear of COVID-19; third, all mental health outcomes of the general population tended to improve from T1 to T2; fourth, the difference in depressive, anxiety, and trauma symptoms over time

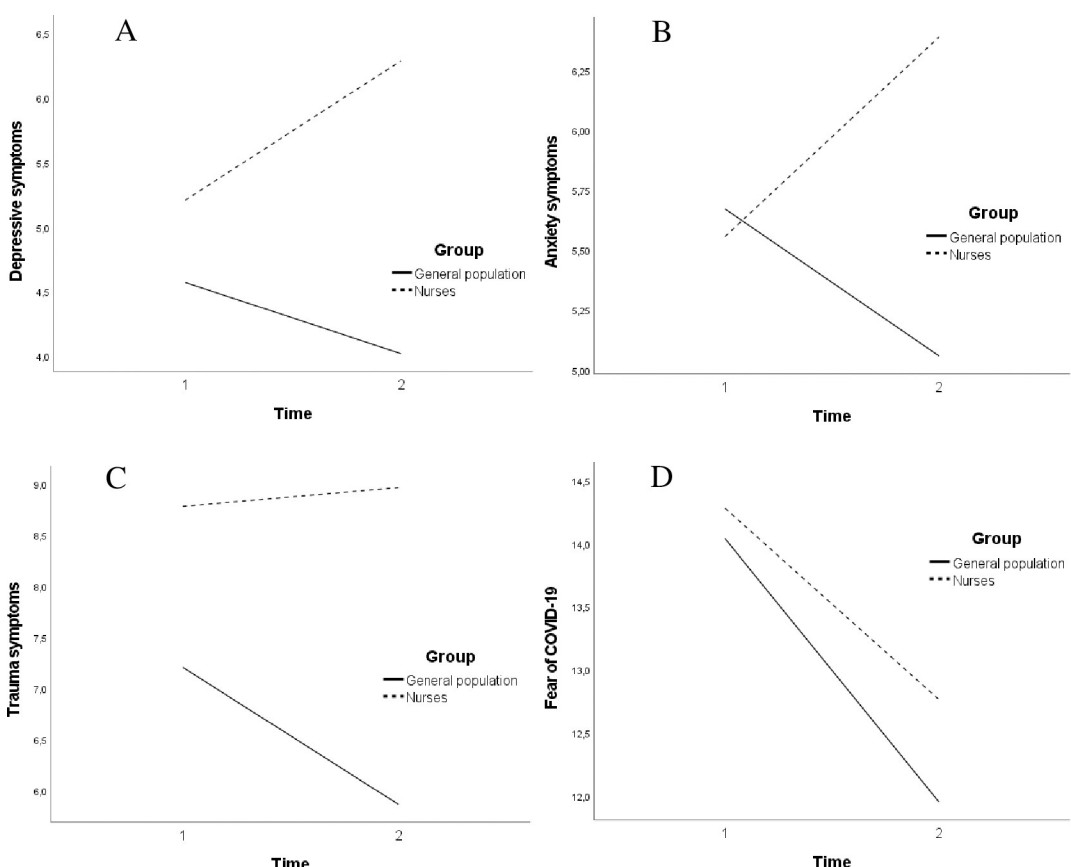

**Fig 1.** Depressive (A), anxiety (B), trauma (C) symptoms, and fear of COVID-19 (D) at T1 and T2 across groups.

was significant between nurses and the general population; and finally, fear of COVID-19 significantly decreased over time for both groups, but especially for the general population.

Nurses' levels of depressive and trauma symptoms, as well as of fear of COVID-19, were greater than those of the general population at every assessment. Moreover, the results revealed that only depressive and trauma symptoms were significantly different between both groups. When analyzing the clinical levels of depressive, anxiety, trauma, and fear of COVID-19 symptoms between nurses and the general population, it was found that groups only showed a significant difference at the follow-up assessment for depressive, anxiety and trauma symptoms, with nurses exhibiting higher scores. Due to exposure to additional sources of stress (e.g., direct contact with the virus, excessive workload, new workplaces and schedules, insufficient personal protective equipment, concerns about becoming infected or infecting significant others), nurses faced an increased risk of developing psychopathological reactions to the pandemic [49]. Perceived loss of resources (e.g., interpersonal relationships, daily routine, deaths, competencies), physical and mental exhaustion, and the pressure to make moral decisions may justify the increased levels of depressive symptoms in these professionals. Additionally, working with people who are suffering, traumatized, seriously ill, and/or dying with the virus may be associated with different forms of trauma-related stress conditions and compromise nurses' ability to cope with the situation [50, 51]. The traumatic nature of the COVID-19 pandemic can also be related to primary (e.g., perceived threat to well-being inherent to exposure to the virus, COVID-19-related uncertainty) and secondary stressors (e.g., lack of social support, family-related concerns, inadequate working conditions) that can cause a combined,

multilayered, complex, and ongoing impact on daily life [1]. Therefore, nurses face different types of stressors, not only in their professional and personal lives but also through their patients' experience (e.g., vicarious traumatization or secondary traumatic stress), which make them more prone to develop a myriad of symptoms, including clinically significant symptomatology.

The general population only demonstrated higher levels of symptoms than nurses at the baseline assessment of anxiety. In fact, anxiety is characterized by a major concern about self-preservation and survival, which is one of the greatest consequences of coronavirus disease. The restrictions in social and professional activities rapidly imposed to control the spread of the virus for an unknown period, as well as the COVID-19-related uncertainty, triggered heightened sensitivity to environmental threat-related stimuli. Individuals were hypervigilant and had the perception of being in constant danger, which was enhanced by the implementation of physical distancing, self-isolation, and preventive health behaviors (e.g., handwashing, mask-wearing). Increasing unemployment rates, as well as the economic recession and financial insecurity that followed, contributed to extreme levels of stress and anxiety. Furthermore, COVID-19 misinformation and disinformation, labelled as infodemic by the WHO, overloaded the general public with false beliefs about the disease, diagnosis, prevention methods, and treatments [52, 53]. Fake news and sensationalized communication about the pandemic, through traditional (e.g., television, radio) and social media (e.g., Facebook, Twitter), might have contributed to amplification of risk perception and feelings of disorientation and lack of control [54]. Given that nurses have higher health literacy, they might be better able to navigate the "infodemic", obtaining more reliable information about the course of the pandemic and its effects, adopting a critical posture about misinformation and being able to integrate the event more easily [55, 56].

Concerning the trajectory of adaptation, nurses showed a worsening in psychological symptoms from the baseline to the follow-up assessment, except for fear of COVID-19, which showed an improvement. These results attest to the long-term effects of the pandemic on the mental health of professionals who had to fight against an invisible and unknown threat, both at work and at home [57, 58]. As the waves of infection passed by, the burden of health care workers increased, with no time to regain energy and process traumatic material. However, the symptoms of the general population declined with the lifting of the lockdown measures and adaptation to the circumstances imposed by the pandemic. These findings were in line with previous longitudinal studies in community samples that found a significant increase in mental health outcomes during the initial stages of the pandemic, followed by a significant decline to baseline scores [14, 59]. For the analyses of symptoms with clinical significance, differences between timepoints were found for both groups on all measures, which means that nurses and the general population demonstrated a significant change from T1 to T2 in the intensity of symptoms.

Furthermore, differences between nurses and the general population on depressive, anxiety, and trauma symptoms from T1 to T2 were significant and moderate, which highlights the specific challenges faced by both groups due to the pandemic outbreak and the subsequent waves of infection. Although the new circumstances imposed strong precautions worldwide, research has demonstrated that there are groups more affected than others, emphasizing the role of social determinants of health [8].

Finally, the levels of fear of COVID-19 significantly decreased from T1 to T2 for both subsamples, but the difference was greater for the general population. Immediately after the COVID-19 outbreak, the knowledge about the disease was extremely limited (i.e., detection, course, effects, treatment), as well as the groups that were more affected and why [60]. The governments of various countries implemented unprecedented public health measures to

contend with SARS-CoV-2 transmission (e.g., border management, school closures, social contact tracing). These political actions, along with fear-inducing messages on mass media, may have amplified perceived threat and elicited extreme levels of public fear, motivating emotion-driven behaviors to protect one's survival (e.g., stockpiling food, toilet paper). Fear experiences during the COVID-19 pandemic can be conceptualized into four interrelated dialectical domains that represent the bodily, interpersonal, cognitive, and behavioral features of fear [61]. Over time, knowledge about the disease improved, vaccines became available, the number of cases and deaths declined, and the restrictions were lifted. People received better information about the transmission process and could adopt adequate measures to cope with it, acting consciously and responsibly to maintain safety without compromising daily living. Continuous exposure to virus-related information through different sources may have also fostered desensitization to those stimuli. Cognitive, emotional, and physiological responses to threat may have declined, and individuals experienced decreased fear of the pandemic [62].

### Limitations and future directions

Despite the contributions of the present research, there are some limitations that should be recognized. First, dropout rates may have influenced the results on the trajectory of psychological symptoms. Although attrition is expected in longitudinal studies, especially due to "research fatigue" in postdisaster research, the differences found between those who stayed in the study and those who dropped out might not be associated with the relationship between variables [32]. However, the literature shows that estimates of associations between variables do not seem to be affected by attrition rate when it is dependent on follow-up variables, which may support the generalizability of the present study. Nevertheless, future studies should address this issue by using more sophisticated techniques to handle missingness and to reduce attrition bias (e.g., full information maximum likelihood and multiple imputation analysis) [33]. Second, mental health outcomes might be influenced by other variables that could explain the differences found over time, as well as between groups. Having underlying medical conditions, living in poor housing settings, or integrating a social minority represented a higher risk for those who got sick from COVID-19, making them more vulnerable to develop acute infection and clinical levels of psychological symptoms [63]. Therefore, the longitudinal effects of the study should be interpreted with caution within the context of the pandemic. In the future, analyses should consider and control variables that might contribute to specific mental health outcomes (e.g., sociodemographic and clinical characteristics, isolation period, being part of the risk group for COVID-19). In addition, it would be valuable to address other adjustment outcomes (e.g., posttraumatic growth, quality of life, obsessive-compulsive symptoms, burnout), seeking to improve a comprehensive model explaining the relationship between health pandemic appraisals and individuals' mental health outcomes. Third, the use of proposed cutoff criteria lacks additional validation for the Portuguese population and across larger samples, which would be valuable for future studies to consider. Fourth, the sample was collected exclusively through online recruitment, which may have excluded participants with the lowest levels of digital literacy or those who do not use social media. Therefore, it is important that data collection complement online recruitment with paper-and-pencil questionnaires. Finally, the specific country and period of the pandemic in which data were collected may limit the generalizability of findings to other cultural contexts and moments of this ever-changing health crisis (in terms of positive COVID-19 cases and associated lockdown measures), thus reinforcing the need to replicate this research for different populations and stages of the pandemic.

## Public health relevance

This study generally showed higher levels of symptoms in all moments of assessment for nurses (excluding anxiety at baseline), as well as a deterioration in their mental health outcomes over time (except for fear of COVID-19). The general population tended to report lower levels of symptoms, demonstrating an improvement in the aftermath of the COVID-19 pandemic. Therefore, there is an urgent need for long-term monitoring of psychosocial effects in the community, but mainly among nurses, who may be more vulnerable in future waves of this pandemic or future health crises. These findings also highlight the importance of tailoring resilience-promoting interventions to the unique needs of this group for a specific context since sociocultural factors (e.g., socioeconomic level, identifying as a marginalized group), as well as the roles endorsed within the community (e.g., leadership roles, support groups), strongly influence the adjustment outcomes in the aftermath of global crises [64, 65]. Nursing professionals are expected to maintain a healthy and thriving community by delivering high-quality care in health care systems. However, the occupational stressors they have faced since the beginning of the COVID-19 outbreak make their jobs harder and more challenging than ever before. These circumstances increase the likelihood of negligent practices, errors, and dehumanized care. Organizational leaders have the responsibility to advocate for the physical and mental well-being of health care staff, even more so during global health crises; they can do so by improving leadership, creating safe and resilient working environments, and promoting social connectedness among peers. Governments and decision-makers also need to acknowledge the importance of creating policies that promote changes at the organizational level and provide adequate support to these professionals so they can in turn support their patients.

## Supporting information

**S1 Checklist. STROBE statement—checklist of items that should be included in reports of observational studies.**
(DOCX)

## Author Contributions

**Conceptualization:** Catarina Vitorino, Maria Cristina Canavarro, Carlos Carona.

**Data curation:** Catarina Vitorino, Carlos Carona.

**Formal analysis:** Catarina Vitorino, Carlos Carona.

**Funding acquisition:** Catarina Vitorino, Carlos Carona.

**Investigation:** Catarina Vitorino, Carlos Carona.

**Methodology:** Catarina Vitorino, Carlos Carona.

**Project administration:** Catarina Vitorino.

**Supervision:** Maria Cristina Canavarro.

**Validation:** Maria Cristina Canavarro.

**Writing – original draft:** Catarina Vitorino.

**Writing – review & editing:** Carlos Carona.

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
