## [Decision Letter · Decision Letter 0]

11 Jan 2024

PONE-D-23-17263Six-month psychopathological symptom trajectories following the COVID-19 outbreak: Contrasting mental health outcomes between nurses and the general populationPLOS ONE

Dear Dr. Vitorino,

Thank you for submitting your manuscript to PLOS ONE. After careful consideration, we feel that it has merit but does not fully meet PLOS ONE’s publication criteria as it currently stands. Therefore, we invite you to submit a revised version of the manuscript that addresses the points raised during the review process.

The reviewers raised small points that need adjustments/clarifications

We look forward to receiving your revised manuscript.

Kind regards,

Flávia L. Osório, PhD

Academic Editor

PLOS ONE

Journal Requirements:

2. In this instance it seems there may be acceptable restrictions in place that prevent the public sharing of your minimal data. However, in line with our goal of ensuring long-term data availability to all interested researchers, PLOS’ Data Policy states that authors cannot be the sole named individuals responsible for ensuring data access (http://journals.plos.org/plosone/s/data-availability#loc-acceptable-data-sharing-methods).

Reviewers' comments:

Reviewer's Responses to Questions

**Comments to the Author**

1. Is the manuscript technically sound, and do the data support the conclusions?

Reviewer #1: Yes

Reviewer #2: Yes

2. Has the statistical analysis been performed appropriately and rigorously? 

Reviewer #1: Yes

Reviewer #2: Yes

3. Have the authors made all data underlying the findings in their manuscript fully available?

Reviewer #1: Yes

Reviewer #2: No

4. Is the manuscript presented in an intelligible fashion and written in standard English?

Reviewer #1: Yes

Reviewer #2: Yes

5. Review Comments to the Author

Reviewer #1: This is a 6-month longitudinal cohort study comparing the questionnaire responses of a sample of 180 nurses to those from a sample of 158 persons in the general population during the Covid pandemic. Questions covered issues of depressive, anxiety, trauma and fear of Covid-19 symptoms. The objective is to evaluate the stability or change in adaptation outcomes between these groups at two time points separated by 6 months.

Comment

It would be good to know the power of the study as it would add to the validity of the study.

The attrition rate of 75% between T1 and T2 is high and this would affect the validity of the study and introduce bias into the study. So have there been attempts to use the "intention to treat" protocol to account for the attrition and has this been planned for in the planning phase of the study?

On data presentation, data presented in the tables need not be repeated in the text but significant points should be highlighted in the text.

You have highlighted some of the biases and shortcomings of the study.

In Fig 1, the slopes of the lines for depression, anxiety and trauma were divergent between the nursing group and the general population. This suggests that over time, the symptoms continued to rise in the nursing group in contrast to the general population where the symptoms tended to decline over time, between T1 and T2. Fear of the pandemic tended to decline over time for both groups

Reviewer #2: The manuscript entitled "Six-month psychopathological symptom trajectories following the COVID-19 outbreak: Contrasting mental health outcomes between nurses and the general population” is well organized and written clearly.

The main goals of the paper are clear, and the research is relevant to the field of health psychology, specifically to the study of the impact of COVID-19 outbreak on health professionals. The research has a longitudinal design, which makes the results more valid and meaningful.

The abstract is well structured and summarizes the methodology and results of the study.

The authors did a good review of previous literature to support their research.

Regarding the Method, the description of the characteristics of the general population sample could be enriched so that it was clearer who the participants were (e.g., professional status or job), also because sample was collected through online recruitment.

In Measures, the authors of the scales “ODSIS”, “OASIS”, “IES-6” and “The Fear of COVID-19 Scale” should be specified in the text.

The data analysis is in line with the research goals. Nevertheless, the analysis of the variable Risk group for COVID-19, that was ignored in the results, could provide interesting findings.

The discussion of the results is appropriate and based on recent studies. The authors report the limitations of the study and its implications.

6. PLOS authors have the option to publish the peer review history of their article (what does this mean?). If published, this will include your full peer review and any attached files.

Reviewer #1: **Yes: **Yuke Tien, Fong

Reviewer #2: No

---

## [Author Response · Author response to Decision Letter 0]

1 Mar 2024

Reviewer #1

This is a 6-month longitudinal cohort study comparing the questionnaire responses of a sample of 180 nurses to those from a sample of 158 persons in the general population during the Covid pandemic. Questions covered issues of depressive, anxiety, trauma and fear of Covid-19 symptoms. The objective is to evaluate the stability or change in adaptation outcomes between these groups at two time points separated by 6 months.

Comment

It would be good to know the power of the study as it would add to the validity of the study.

The attrition rate of 75% between T1 and T2 is high and this would affect the validity of the study and introduce bias into the study. So have there been attempts to use the "intention to treat" protocol to account for the attrition and has this been planned for in the planning phase of the study?

• Thank you for raising this point. We do endorse your concern. The influence of the attrition rate on the results is depicted in the Limitations section, especially due to “research fatigue” in the aftermath of a mass traumatic event such as the COVID-19 pandemic (Patel et al. 2020). However, the literature suggests that the intention-to-treat protocol usually applies to randomized clinical trials (Fisher et al, 1990). According to your note, we tried to clarify this topic (“However, the literature shows that estimates of associations between variables do not seem to be affected by attrition rate when it is dependent on follow-up variables, which may support the generalizability of the present study. Nevertheless, future studies should address this issue by using more sophisticated analysis to handle missing values and reduce bias (e.g., multiple imputation analysis) (Gustavson et al., 2012).”).

On data presentation, data presented in the tables need not be repeated in the text but significant points should be highlighted in the text.

• As you recommended, information presented in tables were better described in the text (e.g., “Regarding the clinical levels of symptoms presented in Table 2, nurses reported higher levels than the general population for all measures, ranging between 22.8% and 31.7% at T1, and 21.7% and 43.3% at T2. Among the general population, symptoms with clinical significance ranged between 15.8% and 29.1% at T1, 11.4% and 24.7% at T2.”)

You have highlighted some of the biases and shortcomings of the study.

• Thank you for acknowledging our best efforts to provide to the PloS One‘s readers with a clear, and reliable research outcome.

In Fig 1, the slopes of the lines for depression, anxiety and trauma were divergent between the nursing group and the general population. This suggests that over time, the symptoms continued to rise in the nursing group in contrast to the general population where the symptoms tended to decline over time, between T1 and T2. Fear of the pandemic tended to decline over time for both groups.

• Thank you for sharing your insights on the analysis of these results, which are very much aligned with the comments that we have elaborated in the Discussion section: “The main findings may be summarized as follows: first, nurses generally reported higher levels of symptoms than the general population; second, the mental health outcomes of nurses tended to deteriorate over time, except for fear of COVID-19; third, all mental health outcomes of the general population tended to improve from T1 to T2; fourth, the difference in depressive, anxiety, and trauma symptoms over time was significant between nurses and the general population; and finally, fear of COVID-19 significantly decreased over time for both groups, but especially for the general population.”.

Reviewer #2

The manuscript entitled "Six-month psychopathological symptom trajectories following the COVID-19 outbreak: Contrasting mental health outcomes between nurses and the general population” is well organized and written clearly.

The main goals of the paper are clear, and the research is relevant to the field of health psychology, specifically to the study of the impact of COVID-19 outbreak on health professionals. The research has a longitudinal design, which makes the results more valid and meaningful.

The abstract is well structured and summarizes the methodology and results of the study.

The authors did a good review of previous literature to support their research.

Regarding the Method, the description of the characteristics of the general population sample could be enriched so that it was clearer who the participants were (e.g., professional status or job), also because sample was collected through online recruitment.

• Thank you so much for your careful and comprehensive review. The general population was better described in the Results section (“As for the general population, most individuals had an academic degree (47.5%) and worked on-site (53.2%).”) with additional information that was collected in the sociodemographic questionnaire.

In Measures, the authors of the scales “ODSIS”, “OASIS”, “IES-6” and “The Fear of COVID-19 Scale” should be specified in the text.

• The authors of the scales were identified in the description of each one of them (please check subsection Measures, section “Methods”).

The data analysis is in line with the research goals. Nevertheless, the analysis of the variable Risk group for COVID-19, that was ignored in the results, could provide interesting findings.

• Thank you for your note. As you mention, being part of the risk group may have a strong effect on mental health outcomes. Thus, the importance of considering this factor when interpreting the results of the longitudinal impact of the pandemic was further detailed in the revised version of the manuscript (“Having underlying medical conditions, living in poor housing settings, or integrating a social minority represented a higher risk for those who got sick from COVID-19, making them more vulnerable to develop acute infection and clinical levels of psychological symptoms [65]. Therefore, the longitudinal effects of the study should be interpreted with caution within the context of the pandemic.”).

The discussion of the results is appropriate and based on recent studies. The authors report the limitations of the study and its implications.

• Thank you for recognizing our efforts in contributing with an informative and honest submission.

---

## [Editor Report · Decision Letter 1]

18 Mar 2024

Six-month psychopathological symptom trajectories following the COVID-19 outbreak: Contrasting mental health outcomes between nurses and the general population

PONE-D-23-17263R1

Dear Dr. Vitorino

We’re pleased to inform you that your manuscript has been judged scientifically suitable for publication and will be formally accepted for publication once it meets all outstanding technical requirements.

Kind regards,

Flávia L. Osório, PhD

Academic Editor

PLOS ONE
---

## [Editor Report · Acceptance letter]

3 Apr 2024

PONE-D-23-17263R1 

PLOS ONE

Dear Dr. Vitorino, 

I'm pleased to inform you that your manuscript has been deemed suitable for publication in PLOS ONE. Congratulations! Your manuscript is now being handed over to our production team.

Kind regards, 

on behalf of

Dr. Flávia L. Osório 

Academic Editor

PLOS ONE